# Presence of CD44v9-Expressing Cancer Stem Cells in Circulating Tumor Cells and Effects of Carcinoembryonic Antigen Levels on the Prognosis of Colorectal Cancer

**DOI:** 10.3390/cancers16081556

**Published:** 2024-04-19

**Authors:** Katsuji Sawai, Takanori Goi, Youhei Kimura, Kenji Koneri

**Affiliations:** First Department of Surgery, University of Fukui, Fukui 910-1193, Japan; tgoi@u-fukui.ac.jp (T.G.); y-kimura@kimura-hospital.jp (Y.K.); koneri@u-fukui.ac.jp (K.K.)

**Keywords:** colorectal adenocarcinoma, CD44 variant exon 9, cancer stem cells, circulating tumor cells, tumor marker

## Abstract

**Simple Summary:**

Circulating tumor cells (CTCs) are cancer cells released from the primary tumor into the bloodstream, and contain cancer stem cells. The present study demonstrates the significance of a specific variant, CD44v9, in CTCs, and its combined effects with preoperative carcinoembryonic antigen (CEA) values on the prognosis of colorectal cancer (CRC). Analysis of the serum CEA levels and the expression of CD44v9 mRNA in CTCs, followed by evaluation of their association with clinicopathological factors, showed the association of CD44v9 mRNA with liver metastasis. Furthermore, patients with CD44v9-positive CTCs showed a poorer prognosis than those with CD44v9-negative CTCs. Combining CD44v9 mRNA expression with CEA values provided more detailed prognostic information. These results suggest that CD44v9 mRNA expression in CTCs, alongside CEA levels, could serve as a valuable prognostic marker for CRC, and could potentially lead to the development of more personalized treatment strategies against CRC.

**Abstract:**

Circulating tumor cells (CTCs) are cancer cells released from the primary tumor into the bloodstream, and contain cancer stem cells that influence tumor survival, recurrence, and metastasis. Here, we investigated CD44v9 expression in CTCs and impact of preoperative carcinoembryonic antigen (CEA) levels on colorectal cancer (CRC) prognosis. We analyzed the expression of CD44v9 mRNA in CTCs using reverse transcription-polymerase chain reaction and preoperative CEA levels in blood samples obtained from 300 patients with CRC. Subsequently, we evaluated the association of CD44v9 expression and CEA levels with clinicopathological factors. CD44v9 mRNA was expressed in 31.3% of the patients, and was significantly associated with liver metastasis. Patients with positive CD44v9 expression had a lower 5-year survival rate (62.3%) than those with negative CD44v9 expression (82.8%, *p* < 0.001). Cox regression analysis identified CD44v9 expression and high CEA levels (≥5 ng/mL) as poor prognostic factors, while negative CD44v9 expression and low CEA levels (<5 ng/mL) were associated with favorable prognosis (hazard ratio = 0.285, *p* = 0.006). These results suggest that a combination of CD44v9 mRNA expression in CTCs and serum CEA levels could serve as a valuable prognostic marker for CRC, potentially enhancing the accuracy of prognosis predictions.

## 1. Introduction

The prevalence of colorectal cancer (CRC) has been increasing in recent years as a major global health challenge, with more than 1.8 million new cases and 881,000 deaths reported worldwide in 2018 [1]. Significant advancements have occurred in the treatment of CRC, especially progress in molecularly targeted therapies and immunotherapy, which have led to improved survival rates. However, identifying the prognostic factors remains crucial for early detection of CRC recurrence and determination of efficacy to enhance treatment outcomes [2]. Carcinoembryonic antigen (CEA), a tumor marker for CRC, recommended by the National Comprehensive Cancer Network guidelines, is widely used in clinical practice for monitoring CRC [3,4,5,6,7,8]. Elevated levels of preoperative CEA levels have been associated with poor prognosis [9]. However, its practical applicability is limited in patients with normal preoperative CEA values. Moreover, the cutoff values, sensitivity, and specificity in determining treatment efficacy vary in different reports, which limits using CEA levels as a sole indicator for monitoring treatment efficacy [10,11,12,13].

Circulating tumor cells (CTCs) are tumor cells that are shed from the primary or metastatic lesions and enter the peripheral bloodstream [14]. The presence of a high number of circulating tumor cells has been suggested to be significantly associated with poor progression-free survival and OS in CRC [15,16,17]. Additionally, in a study involving 430 patients with metastatic CRC, the number of CTCs before and after treatment has been identified as an independent predictor of OS [18]. They also include cancer stem cells that exhibit a self-renewal potential and resistance to anticancer drugs [19,20,21,22]. Tumors comprise a heterogeneous population of cells that exhibit or lack the ability to self-renew. Among these, cells with the ability to self-renew are known as cancer stem cells or tumor-initiating cells, which are thought to be responsible for initiating and driving tumor growth. These cells are also involved in resistance to chemotherapy, as well as in the differentiation process, through which they produce many cancer cells that lack tumor-forming ability [23,24,25,26]. Therefore, therapies targeting cancer stem cells are considered promising to prevent tumor relapse and metastasis. In CRC, several cancer stem cell-specific markers have been identified, including leucine-rich repeat-containing G protein-coupled receptor 5, CD133, and CD44 [27,28,29]. A previous study has reported that the presence of *CD133* mRNA expression in CTCs is associated with poor prognosis of CRC [30,31].

CD44 functions as a single-pass type I transmembrane protein that acts as a cell adhesion molecule to hyaluronic acid, a major component of the extracellular matrix. CD44 is involved in various physiological processes, including leukocyte homing and activation, wound healing, and cell migration [32].

CD44 is encoded by a single gene with 19 exons. The first five and the last five exons are constant, coding for the shortest isoform of CD44 with a molecular weight of 85–95 kDa, known as standard CD44 (CD44s). Variant isoforms of CD44 (CD44v) are generated through alternative splicing, resulting in a combination of the ten constant exons with any combination of the remaining nine variant exons [33,34,35,36]. Each isoform has distinct functions, and among them, CD44v8-10 binds to xCT protein on the cell membrane, which transports cystine/glutamate. This binding promotes the formation of reduced glutathione, inhibits the accumulation of reactive oxygen species in cancer cells, and suppresses the activation of oxidative stress, thereby conferring self-renewal capabilities and resistance to chemotherapy [37]. Oxidative stress has been implicated in the development, proliferation, and metastasis of colorectal cancer. We have reported that patients with colorectal cancer who have high levels of serum oxidative stress have a poor prognosis [38]. The expression of CD44v9, which has the ability to control oxidative stress, may be an effective marker for prognosis prediction, and may be a target for antibody therapy.

CD44v9, one of the splicing variants of CD44, is closely associated with tumorigenicity and several cellular processes, including cell proliferation, metastasis, and tumor invasiveness through epithelial-mesenchymal transition (EMT) [25]. A previous study has shown that CD44v9 siRNA-treated cancer cells exhibit increased oxidative stress upon exposure to 5-fluorouracil, compared to the untreated control cells [39]. Furthermore, inhibition of CD44v9 was identified as a promising strategy for developing treatment strategies for CRC, with potential implications for the development of therapeutic drugs [25,39].

In a study involving 193 patients with gastric cancer, the expression of CD44v9 detected by immunostaining was shown to be significantly associated with the depth of cancer invasion, lymphatic invasion, vascular invasion, distant metastasis, and the expression of GPx2 [39]. The study also identified CD44v9 as an independent prognostic factor for poor overall (OS) and recurrence-free survival [39]. Furthermore, CD44v9 expression in tumor tissue is considered a poor prognostic factor in several carcinomas, including colorectal and hepatocellular carcinoma [32,39,40,41,42]. However, reports on the characteristics and prognosis of CRC cases with CD44v9-expressing CTCs are limited. Additionally, the number of cases studied is limited, and a comprehensive approach that considers the combined effect of CD44v9-positive CTCs and preoperative CEA values is lacking. Studying both CEA levels and CD44v9-expressing CTCs together could provide a more comprehensive assessment of CRC prognosis. While elevated CEA levels may indicate a general risk of disease progression, the presence of CD44v9-expressing CTCs could identify a subset of patients with a particularly aggressive cancer phenotype. By combining these biomarkers, clinicians may be able to better stratify patients based on their risk profiles, and tailor treatment strategies accordingly, ultimately improving treatment outcomes in CRC. Therefore, in this study, we aimed to investigate the prognostic effects of the presence of CD44v9-positive CTCs and preoperative CEA levels and confirm their validity as biomarkers.

## 2. Materials and Methods

### 2.1. Patients and Sample Collection

Three hundred patients who underwent CRC resection at our institution between 2013 and 2018 were enrolled in this study. Patients with synchronous or metachronous cancers were excluded. In addition, 15 healthy donors were included as controls.

Blood samples from the patients were collected before primary tumor resection to measure the preoperative CEA levels in CTCs with or without CD44v9 expression. The normal reference value for CEA was 5 ng/mL. The first 5 mL of blood was discarded to minimize the risk of skin cell contamination. Then, 20 mL of blood was collected and separated using an OncoQuick density gradient system (Greiner Bio-One GmbH, Frickenhausen, Germany) following the manufacturer’s instructions, and the tumor cells were isolated through density gradient centrifugation. Subsequently, the tumor cells were resuspended in 400 μL of phosphate-buffered saline. For a negative control, blood (without epithelial cells) from healthy volunteers was collected using the OncoQuick density gradient system.

The Research Ethics Committee of the University of Fukui approved the study (Approval No. 20200058). Written informed consent was obtained from the patients for publication of this research project.

### 2.2. Reverse Transcription-Polymerase Chain Reaction (RT-PCR)

Total RNA was isolated from tumor cells using the ISOGEN (Wako, Tokyo, Japan) and was reverse transcribed using a Prime Script RT reagent kit (Takara, Otsu, Japan). The coding regions of *CD44v9* were amplified using the following primers: forward: AGCAGAGTAATTCTCAGAGC and reverse: TGATGTCAGAGTAGAAGTTGTT [43]. The thermal cycling conditions comprised 35 cycles of denaturation at 94 °C for 1 min, annealing at 55 °C for 1 min, and extension at 72 °C for 2 min. The amplification was performed using a PTC-100 Programmable Thermal Controller (NJ Research Inc., Manahawkin, NJ, USA). The PCR products were then purified using the QIAquick PCR Purification kit (Qiagen, Hilden, Germany) and analyzed using gel electrophoresis (1.2% agarose). The purified PCR products were then sequenced to confirm the presence of *CD44v9*.

For semi-quantitative mRNA detection, ethidium bromide staining was performed to identify *CD44v9* bands in the gels. To ensure consistent results, all PCR amplifications were duplicated. The amplicons in photographed gels were quantified using densitometry.

### 2.3. Clinical Assessment

Data on patient demographics (age, sex), tumor characteristics (size, location, histological type, invasion depth), metastasis status (lymph node and distant metastasis), cancer stage, CEA levels, and disease-specific survival (DSS) were obtained. DSS was calculated as the time from the date of surgery to death, specifically from CRC. Histopathological and clinical staging of tumors were based on the TNM classification system.

All patients underwent follow-up assessments, including blood tests for tumor markers every 3 months, enhanced abdominal computed tomography every 6 months, and colonoscopy every 3 years.

### 2.4. Statistical Analysis

The Kaplan–Meier method was used to analyze DSS, and comparisons between groups were performed using a log-rank test. The Cox regression model was used to assess the hazard ratio (HR). Other characteristics of the two groups were compared using the chi-square test for univariate analysis and logistic regression analysis for multivariate analysis. All statistical analyses were performed using IBM SPSS software version 21.0 (IBM Japan, Ltd., Tokyo, Japan). Differences were considered significant at *p* < 0.05.

## 3. Results

### 3.1. Association between CD44v9 Expression and Clinicopathologic Features

Baseline demographic and clinicopathologic data of all patients with CRC are summarized in Table 1. The median age of the patients was 69.5 years (range: 24–91 years). Among the 300 patients, 72, 82, 85, and 61 belonged to stages I, II, III, and IV of CRC, respectively. *CD44v9* was expressed in 94 of 300 cases (31.3%; Figure 1), whereas none of the healthy donors expressed *CD44v9* mRNA. Stage-wise, *CD44v9* mRNA expression was positive in 19 cases with stage I, 21 cases with stage II, 24 cases with stage III, and 30 cases with stage IV CRC (Figure 2). Univariate analysis revealed no correlation between *CD44v9* expression and tumor size, tissue differentiation, or depth of disease, whereas it identified a significantly positive correlation between *CD44v9* expression and lymph nodes, distant metastases, and advanced-stage disease. No association was observed between serum CEA levels and *CD44v9* expression (Table 1). Multivariate logistic regression analysis revealed that *CD44v9* was significantly expressed in liver metastasis cases (odds ratio (OR) = 2.697, 95% confidence interval (CI) = 1.122–6.481, *p* = 0.027). However, no significant relationship with other pathological characters was observed (Table 2).

### 3.2. Association between Expression of CD44v9 mRNA in CTCs and Survival Rate

The 5-year survival rate of *CD44v9*-positive cases was 62.3%, and that of *CD44v9*-negative cases was 82.8% (Figure 3). This finding indicated that *CD44v9*-positive cases had a significantly worse prognosis (*p* < 0.001). However, no significant relationship was observed between *CD44v9* expression and prognosis in stage I (Appendix A). In stage II CRC cases, the 5-year DSS rate was 86.4% for *CD44v9*-positive cases and 100% for *CD44v9*-negative cases (Appendix A). Similarly, in patients with stage III CRC, the 5-year DSS rate was 66.2% and 87.8% for *CD44v9*-positive and -negative cases, respectively (Appendix A). In stage IV CRC cases, the 2-year DSS for *CD44v9*-positive cases was 54.1%, whereas that for *CD44v9*-negative was 100% (Appendix A). These findings indicated a significantly worse prognosis for *CD44v9*-positive cases in stages II (*p* = 0.039), III (*p* = 0.038), and IV (*p* = 0.028).

### 3.3. Association between CEA Values and Survival Rates

A total of 122 cases with CEA ≥ 5 ng/mL (high), including 12, 30, 36, and 44 cases in stages I, II, III, and IV, respectively, were identified. The 5-year survival rate for cases with high CEA was 60.0%, whereas it was 87.1% for cases with CEA < 5 ng/mL (low), indicating a significantly poorer prognosis for cases with high CEA levels (*p* < 0.001; Figure 4). Correlation analysis revealed no significant correlation between the CEA levels and the prognosis of patients with stages I and II CRC (Appendix A). However, for stage III cases, the 5-year DSS rate was 71.5% for those with high CEA and 88.9% for cases with low CEA (Appendix A). Similarly, the 2-year DSS rate of patients with stage IV CRC with high CEA was 51.5%, while it was 82.4% for those with low CEA (Appendix A). These results indicate that increased CEA level is a worse prognostic factor for stage III (*p* = 0.041) and IV cases (*p* = 0.031).

### 3.4. Multivariate Cox Analysis for DSS

Table 3 shows the results of multivariate Cox analysis of age, gender, tumor size, serosa invasion, lymph node metastasis, distant metastasis, expression of *CD44v9*, and CEA value for DSS in all patients. The results revealed a significant association between DSS and lymph node metastasis, distant metastasis, *CD44v9* expression, and CEA value (*p* < 0.05 in all cases).

### 3.5. Comparison of DSS between Cases with Negative CD44v9 mRNA Expression and CEA Level

Of all cases, there was negative CD44v9 mRNA expression and low CEA in 122 cases, negative CD44v9 mRNA expression and high CEA in 84 cases, positive CD44v9 mRNA expression and low CEA in 55 cases, positive CD44v9 mRNA expression and low CEA in were 39 cases, and positive CD44v9 mRNA expression and low CEA were in 55 cases and 39 cases, respectively. The 5-year survival rates were 43.1%, 76.9%, 69.5%, and 96.1%, respectively. There was a significant difference in survival rates between groups (P < 0.001; Figure 5).

### 3.6. Comparison of DSS between Cases with Negative CD44v9 mRNA Expression and Low CEA < 5 ng/mL) and Other Cases

The number of cases with different stages of CRC with negative *CD44v9* mRNA expression and low CEA was 121, including 42 of 72 cases with stage I CRC, 35 of 82 cases with stage II CRC, 36 of 85 cases with stage III, and 8 of 61 cases with stage IV CRC. The 5-year DSS of cases with negative *CD44v9* mRNA expression and low CEA did not differ from that of other cases in stages I and II (96.3% vs. 95.2%; *p* = 0.892, and 100% vs. 92.7%; *p* = 0.263, respectively; Appendix A). In contrast, a significantly better prognosis was observed in terms of 5-year DSS in stage III and 2-year DSS in stage IV for cases with negative *CD44v9* mRNA expression and low CEA compared to other cases (100% vs. 67.9%; *p* = 0.001 and 100% vs. 54.1%; *p* = 0.028, respectively; Appendix A). Furthermore, the 5-year survival rates of cases across all stages of CRC with negative *CD44v9* mRNA expression and low CEA were better than those in other cases (92.1% vs. 65.4%, *p* < 0.001; Figure 6). Cox regression analysis revealed that lymph node metastasis, distant metastasis and negative *CD44v9* mRNA expression and low CEA significantly affected prognosis (*p* < 0.05 in all cases; Table 4).

### 3.7. Comparison of DSS between Cases with Positive CD44v9 mRNA Expression and High CEA (≥5 ng/mL) and Other Cases

Positive *CD44v9* mRNA expression and high CEA were observed in 2 out of 72 cases with stage I, 4 out of 82 cases with stage II, 12 out of 85 cases with stage III, and 21 out of 61 cases with stage IV of CRC. No difference was observed in 5-year DSS between cases with positive *CD44v9* expression and high CEA and other cases in stages I (100% vs. 95.6%, *p* = 0.764) and III (72.9% vs. 82.5%, *p* = 0.374) (Appendix A). In cases with stages II and IV exhibiting positive *CD44v9* mRNA expression and high CEA, a significantly worse prognosis was observed in terms of 5-year and 2-year DSS in stages II (50% vs. 98.5% *p* = 0.009) and IV (38.1% vs. 72.1% *p* = 0.007), respectively (Appendix A). In addition, the comparison of 5-year survival rates across all stages of CRC revealed that cases with positive *CD44v9* mRNA expression and high CEA had worse 5-year DSS compared with the other cases (43.1% vs. 81.5%, *p* < 0.001; Figure 7). Cox regression analysis revealed that the presence of lymph node metastasis, the presence of distant metastasis, and positive *CD44v9* mRNA expression and high CEA were the factors significantly associated with worse prognosis (*p* < 0.05 in all cases; Table 5).

## 4. Discussion

In this study, we investigated the correlation between the expression of *CD44v9* mRNA in CTCs and the prognosis of CRC. Our findings revealed that in patients with CRC, the prognosis was poor when CTCs expressed *CD44v9*. We also showed that the expression of *CD44v9* mRNA in CTC was significantly higher in cases with liver metastasis. Finally, we showed that cases with negative *CD44v9* expression in CTCs and normal CEA levels showed significantly better prognoses compared with other cases.

Our study is the first to investigate the association between *CD44v9* mRNA expression in CTCs and the prognosis of CRC using 300 cases. Concordant with studies on other cancer stem cell markers, the *CD44v9* mRNA expression in CTCs was confirmed to be significantly associated with poor prognosis. A few studies have used multivariate analysis to investigate clinicopathological factors affecting the expression of cancer stem cells in CTC. For instance, Chao et al. investigated factors affecting liver metastasis in CRC, and highlighted high CEA levels, extra nodal tumor deposits, and the expression of CD133, CD44, and CD54 in CTCs [44]. CTCs include epithelial tumor cells, tumor cells undergoing EMT, and cancer stem cells, among which cancer stem cells are considered to be involved in metastasis [45,46,47].

The hematogenous metastasis of CRC progresses in the sequence of dissociation of cancer stem cells from the primary focus, invasion into capillaries, metastasis to the whole body through the portal and systemic circulation, adhesion to the vascular endothelial cells of the target organ, extravasation, and infiltration and proliferation [48]. CD44 has been reported to be related to tumor invasion [49], and in gastric and CRCs, proteins expressed from CD44 variant exon 6 or exon 9 have been reported to be involved in hematogenous metastasis [50,51,52]. In this study, the expression of CD44v9 in CTCs was significantly higher in cases with liver metastasis, suggesting that *CD44v9*-expressing cancer stem cells are involved in liver metastasis. Seki used the colorectal cancer cell line HT29, which expresses CD44v9, to develop a highly metastatic cell line. By injecting these cells into the spleens of mice, they were able to create a model of liver metastasis. They reported that liver metastasis formation was inhibited when these HT29 cells were treated with an anti-CD44v9 monoclonal antibody, thus proving that CD44v9 promotes liver metastasis. Additionally, they reported that CD44v9 expression is involved in the adhesion of tumor cells to vascular endothelial cells, a critical factor for metastasis [53]. Circulating tumor cells (CTCs) are tumor cells that are shed from primary or metastatic tumors into the peripheral bloodstream [54], and are frequently detected in stage III and IV blood samples. Cancer stem cells such as CD44 and CD133 have also been identified in tissues from colorectal cancer liver metastases [49], and an increase in CD44v9 expression in liver metastasis cases in this study is presumed to result from shedding from the primary and metastatic sites. Cho et al. reported that the expression of CD44, CD133, and CD54 in CTCs was higher in cases with liver metastasis compared to those without, and that cases exhibiting expression of these markers had a poorer prognosis compared to those that did not [19]. We also investigated the relationship between the expression of CD44v9 in CTCs and CEA. Lin et al. [31] reported a correlation between the expression of *CD133* mRNA and CEA values in univariate analysis [31]. Here, we conducted a multivariate analysis and confirmed no correlation between the expression of *CD44v9* in CTCs and CEA. These findings suggest that the verification of *CD44v9* mRNA expression in CTCs could be considered an effective marker different from existing tumor markers.

Preoperative CEA levels have been shown to be reliable prognostic markers, with higher preoperative CEA levels associated with poorer prognosis [55,56]. In this study, preoperative CEA was also identified as an effective predictor of prognosis. The findings indicated the potency of *CD44v9* mRNA expression and serum CEA levels as CRC biomarkers. Iinuma et al. studied the impact of *CD133* mRNA expression in CTCs on OS and disease-free survival in 735 cases of CRC [57]. The study revealed poor prognosis in the presence of CEA, CK, and *CD133* mRNA. Another study reported that combining CTC count and CEA levels improved the accuracy of prognosis prediction for patients with CRC [58,59]. Therefore, we investigated the possibility of achieving a more accurate prognostic predictor by combining the expression of *CD44v9* mRNA in CTCs with CEA levels.

The analysis of survival involving 300 patients with CRC in this study revealed that the expression of *CD44v9* showed an HR of 1.817 (*p* = 0.029). However, in cases exhibiting *CD44v9* expression with high CEA levels, the HR was 1.771 (*p* = 0.047). Conversely, in cases with negative *CD44v9* expression and low CEA, the HR was 0.285 (*p* = 0.006). These results indicated that evaluating the expression of *CD44v9* alone can sufficiently predict poor prognosis. However, considering the expression of *CD44v9* along with CEA values could be more effective in predicting the prognosis. This strategy can play a supportive role when considering the application of adjuvant chemotherapy in cases with poor general conditions or elderly patients with stage II or III disease. In addition to CEA, which we reported on, Sialyl Lex (SLX) has been reported as a factor related to the expression of CD44v9 that affects the prognosis of colorectal cancer cases. SLX is known to play an important role in the adhesion between tumor cells and endothelial cells. It has been reported that if both are expressed in the immunostaining of colorectal cancer tissues, the prognosis is poor, whereas if neither is expressed, the prognosis is good [60].

Sulfasalazine, used for inflammatory diseases such as ulcerative colitis and Crohn’s disease, is considered to specifically inhibit cystine transport via xCT, thereby selectively controlling the proliferation of cancer cells expressing CD44v [61]. Based on these studies, clinical research is also being conducted, suggesting the potential development of therapies targeting CD44v9 [39,62,63,64].

The study has some limitations. Firstly, the low number of deaths in stages II and III cases made it challenging to perform a multivariate analysis to assess the impact of *CD44v9* expression in CTCs on the prognosis of these cases. However, the univariate analysis indicated that *CD44v9* expression could be a potential risk factor for poor prognosis. Secondly, when examining OS in stage IV cases, it is important to consider that the prognosis for stage IV CRC has significantly improved with the advancements in chemotherapy approaches and the introduction of molecular-targeted drugs.

## 5. Conclusions

In summary, the present study highlights the significance of *CD44v9*-positive CTCs and preoperative CEA levels as prognostic markers in CRC. Furthermore, no correlation between CEA and *CD44v9* mRNA in CTC suggests that these two markers serve as a unique tumor marker, while their combination is more effective in identifying cases with a favorable prognosis.

## Figures and Tables

**Figure 1 cancers-16-01556-f001:**
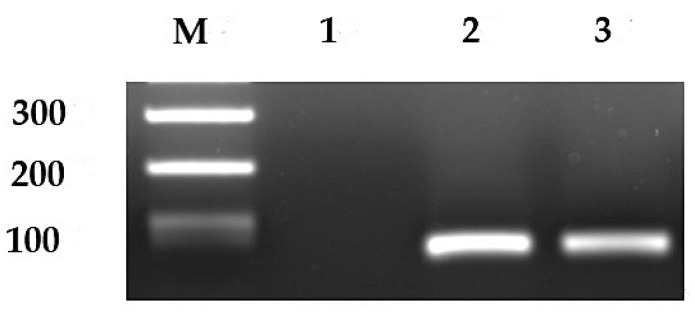
Representative image of expression of *CD44v9* mRNA. Lane 1, negative expression of *CD44v9* mRNA; lanes 2 and 3, positive expression of *CD44v9*; M, DNA size marker. Original western blots are presented in Appendix A.

**Figure 2 cancers-16-01556-f002:**
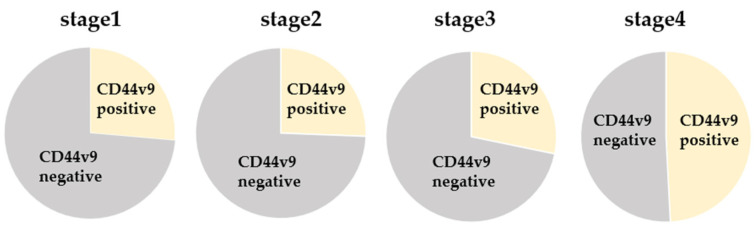
Comparison of CD44v9 mRNA expression by stage.

**Figure 3 cancers-16-01556-f003:**
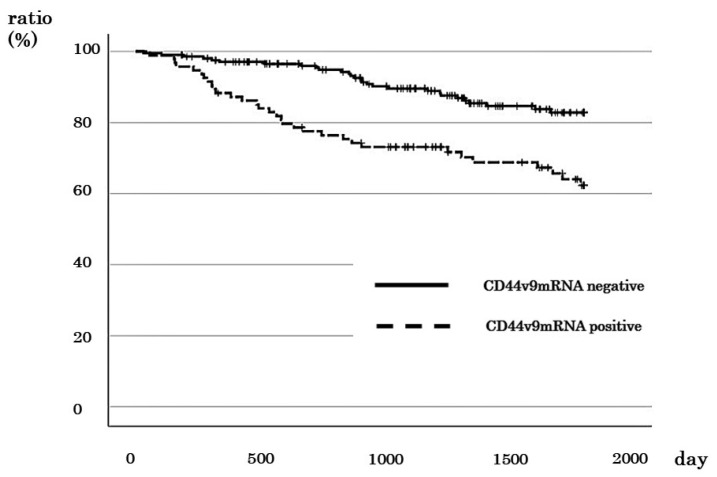
Relationship between *CD44v9* mRNA expression and survival rate in patients with all stages of colorectal cancer.

**Figure 4 cancers-16-01556-f004:**
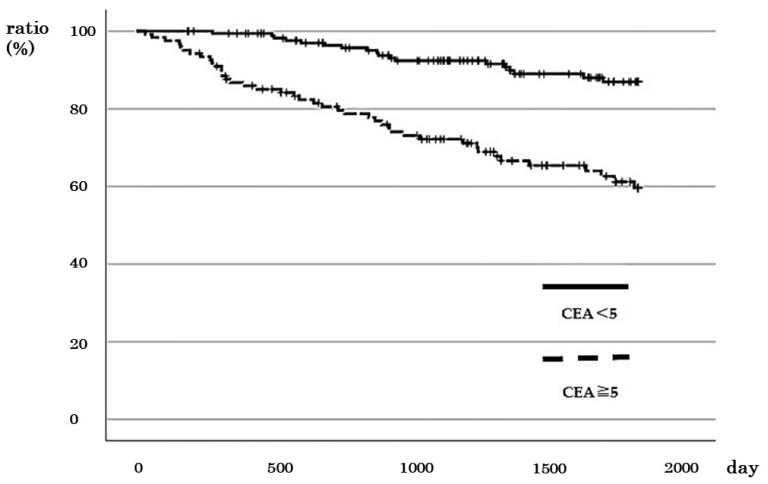
Relationship between CEA values and survival rate in patients with colorectal cancer with all stages.

**Figure 5 cancers-16-01556-f005:**
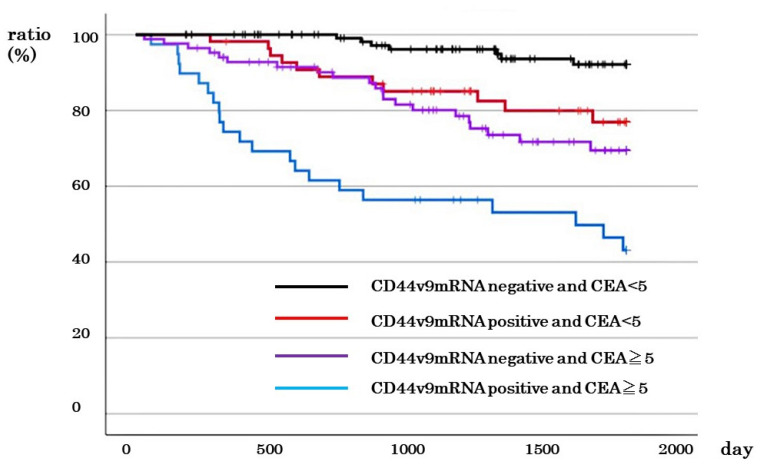
Relationship between *CD44v9* mRNA expression and CEA level and survival rate in patients with all stages of colorectal cancer.

**Figure 6 cancers-16-01556-f006:**
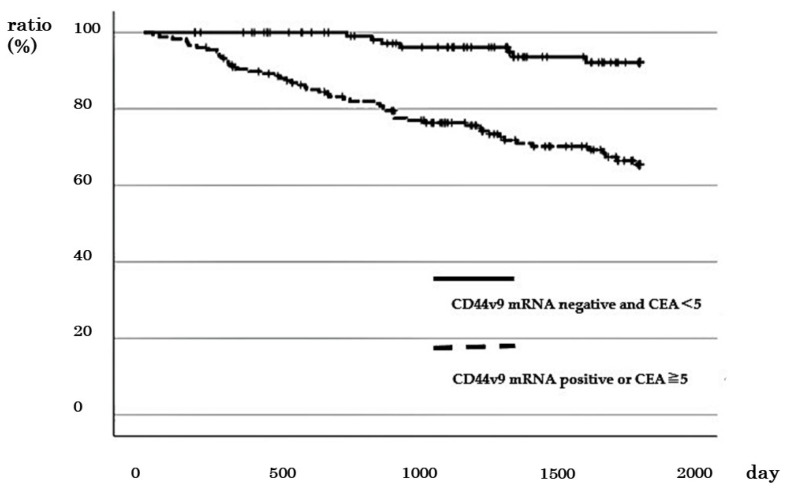
Relationship between negative *CD44v9* mRNA expression and CEA < 5 and survival rate in patients with all stages of colorectal cancer.

**Figure 7 cancers-16-01556-f007:**
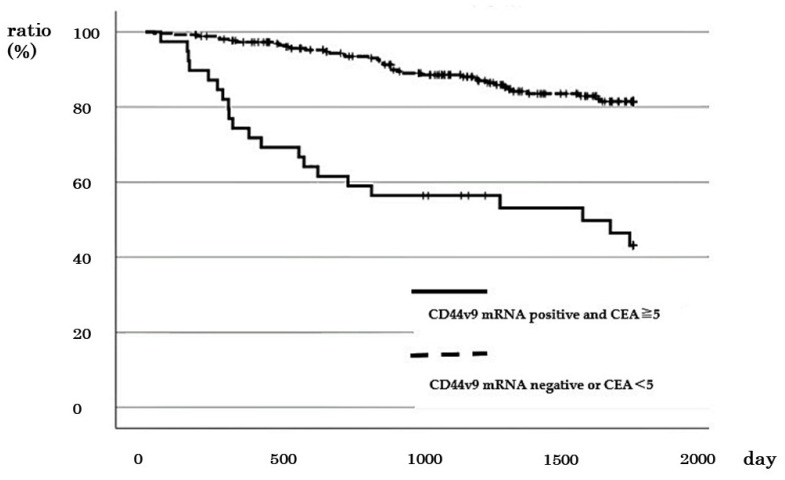
Relationship between positive *CD44v9* mRNA expression and CEA ≥ 5 ng/mL and survival rate patients with all stages of colorectal cancer.

**Table 1 cancers-16-01556-t001:** CD44v9 expression and clinicopathological factors analyzed using univariate analysis.

			*CD44v9* mRNA		
		No. of Cases	Negative Cases (%)	Positive Cases (%)	*p*-Value
All cases (%)		300	206 (68.7)	94 (31.3)	
Age (average years)			66.8	69.5	0.068
Gender	Male	177	122 (68.9)	55 (31.1)	0.907
	Female	123	84 (68.3)	39 (31.7)	
Location	Right colon	110	71 (64.6)	39 (35.5)	0.242
	Left colon	190	135 (71.5)	55 (28.6)	
Size (average mm)			45.08	47.00	0.462
Histological type	Differentiated	278	189 (68.0)	89 (32.0)	0.366
	Undifferentiated	22	17 (77.3)	5 (22.7)	
Serosa invasion	Negative	85	64 (75.3)	21 (24.7)	0.120
	Positive	215	142 (66.0)	73 (34.0)	
Lymph node metastasis	Negative	160	119 (74.4)	41 (25.6)	0.023
	Positive	140	87 (62.1)	53 (37.9)	
Liver metastasis	Negative	263	187 (71.1)	76 (28.9)	0.015
	Positive	37	19 (51.4)	18 (48.6)	
Lung metastasis	Negative	265	189 (71.3)	76 (28.7)	0.043
	Positive	15	7 (46.7)	8 (53.3)	
Stage	I, II	154	114 (74.0)	40 (26.0)	0.040
	III, IV	146	92 (63.0)	54 (37.0)	
CEA	5.0>	172	117 (68.0)	55 (32.0)	0.781
	5.0≤	128	89 (69.5)	39 (30.5)	

**Table 2 cancers-16-01556-t002:** CD44v9 expression and clinicopathological factors analyzed using multivariate analysis.

		Multivariate		
	Variable	Odds Ration	95% CI	*p*-Value
Age (years)		0.977	0.955–1.000	0.48
Gender	Male vs. Female	1.040	0.587–1.843	0.893
Size		0.998	0.982–1.014	0.775
Histological type	Differentiated vs. Undifferentiated	0.545	0.175–1.694	0.294
Serosa invasion	Negative vs. Positive	1.594	0.706–3.599	0.262
Lymph node metastasis	Negative vs. Positive	1.837	0.983–3.433	0.056
Liver metastasis	Negative vs. Positive	2.728	1.131–6.583	0.026
Lung metastasis	Negative vs. Positive	1.544	0.478–4.991	0.468
CEA	5.0> vs. 5.0≤	0.671	0.359–1.255	0.212

**Table 3 cancers-16-01556-t003:** Multivariate Cox analysis for DSS in patients with different stages of CRC.

			Multivariate	
	Variable	HR	95% CI	*p*-Value
Lymph node metastasis	Negative vs. positive	5.430	2.068–14.049	<0.001
Distant metastasis	Negative vs. positive	9.281	4.743–17.582	<0.001
Expression of CD44v9	Negative vs. positive	1.788	1.055–3.001	0.029
CEA value	<5 vs. ≥5	1.888	1.066–3.356	0.030

**Table 4 cancers-16-01556-t004:** Multivariate Cox analysis of different factors, including negative *CD44v9* mRNA expression and CEA < 5 ng/mL for DSS in patients with different stages of CRC.

			Multivariate	
	Variable	HR	95% CI	*p*-Value
Lymph node metastasis	Negative vs. positive	6.143	2.297–16.429	<0.001
Distant metastasis	Negative vs. positive	9.733	5.073–18.672	<0.001
CD44v9 and CEA	CD44v9 (negative) and CEA < 5 vs. others	0.314	0.137–0.717	0.011

**Table 5 cancers-16-01556-t005:** Multivariate Cox analysis tested for various factors, including positive *CD44v9* mRNA expression and CEA ≥ 5 ng/mL for DSS in all patients with different stages of CRC.

			Multivariate	
	Variable	HR	95% CI	*p*-Value
Lymph node metastasis	Negative vs. positive	5.366	2.088–13.793	<0.001
Distant metastasis	Negative vs. positive	9.864	5.160–18.857	<0.001
CD44v9 and CEA	CD44v9 (positive) and CEA ≥ 5 ng/mL vs. others	1.822	1.045–3.177	0.035

## Data Availability

All data included in this study are available upon request by contact with the corresponding author.

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
