# Peer review of "Presence of CD44v9-Expressing Cancer Stem Cells in Circulating Tumor Cells and Effects of Carcinoembryonic Antigen Levels on the Prognosis of Colorectal Cancer"

_cancers, 2024, doi:10.3390/cancers16081556_

Round 1
Reviewer 1 Report
Comments and Suggestions for Authors
The authors have discovered CD44v9 as a potential prognostic marker for CRC. They have looked at patient data and assessed levels of CD44v9 in serum to understand how it correlates with patient data and CRC incidence. I have few suggestions below to make the manuscript better.
1. The introduction needs to be expanded to include few lines on other cancer stem cell markers and their association with tumor growth, survival etc. Some of this is in the discussion and should be moved to the introduction. The discussion is more to talk about implications of their study and its results.
2. The introduction needs to explain rationale for the study. Why did the authors choose this particular CSC marker and not others?
3. The authors need to include some pie charts or venn diagrams to emphasize some of their points. The mRNA levels of CD44v9 are more in stage 4 CRC compared to stage 1 and this can be illustrated in venn diagram or pie chart to better get the point across to readers. This is pretty significant and provides good rationale for their study.
4. In the discussion can authors comment on why are metastatic tumors having more CD44v9 levels in serum? Is it coming from primary tumors or secondary ones? Can authors delineate this?
Author Response
- The introduction needs to be expanded to include few lines on other cancer stem cell markers and their association with tumor growth, survival etc. Some of this is in the discussion and should be moved to the introduction. The discussion is more to talk about implications of their study and its results.
(Response)
We appreciate your helpful suggestions. In response to your comment, we have restructured the lines from 7 to 52 in the Discussion section and incorporated them into the Introduction. This revision includes information about various cancer stem cell markers and their association with tumor growth and survival, providing a broader context at the beginning of our manuscript.
Additionally, we have enriched the Discussion section by adding an analysis on the reasons for the increase in CD44v9 mRNA expression rates in colorectal cancer metastasis cases. We have also included details of new cell experiments that demonstrate the relationship between CD44v9 and liver metastasis.
These changes have been highlighted in yellow in the revised manuscript for ease of review.
- The introduction needs to explain rationale for the study. Why did the authors choose this particular CSC marker and not others?
(Response)
We appreciate your helpful suggestions. Oxidative stress has been implicated in the development, proliferation, and metastasis of colorectal cancer. We have reported that patients with colorectal cancer who have high levels of serum oxidative stress have a poor prognosis. The expression of CD44v9, which has the ability of control oxidative stress, may be an effective marker for prognosis prediction and may be a target for antibody therapy.
The above text has been added to the introduction. These changes have been highlighted in green in the revised manuscript for ease of review.
- The authors need to include some pie charts or venn diagrams to emphasize some of their points. The mRNA levels of CD44v9 are more in stage 4 CRC compared to stage 1 and this can be illustrated in venn diagram or pie chart to better get the point across to readers. This is pretty significant and provides good rationale for their study.
We appreciate the reviewer’s suggestions. Figure 2 shows a pie chart of CD44v9 mRNA expression per Stage.
- In the discussion can authors comment on why are metastatic tumors having more CD44v9 levels in serum? Is it coming from primary tumors or secondary ones? Can authors delineate this?
As suggested, we have added the following text to the discussion
“Goi used the colorectal cancer cell line HT29, which expresses CD44v9, to develop a highly metastatic cell line. By injecting these cells into the spleens of mice, they were able to create a model of liver metastasis. They reported that liver metastasis formation was inhibited when these HT29 cells were treated with an anti-CD44v9 monoclonal antibody, thus proving that CD44v9 promotes liver metastasis. Additionally, they reported that CD44v9 expression is involved in the adhesion of tumor cells to vascular endothelial cells, a critical factor for metastasis. Circulating tumor cells are tumor cells that are shed from primary or metastatic tumors into the peripheral bloodstream, and are frequently detected in stage 3 and 4 blood samples. Cancer stem cells such as CD44 and CD133 have also been identified in tissues from colorectal cancer liver metastases, and an increase in CD44v9 expression in liver metastasis cases in this study is presumed to result from shedding from the primary and metastatic sites. Cho et al. reported that the expression of CD44, CD133, and CD54 in CTCs was higher in cases with liver metastasis compared to those without, and that cases exhibiting expression of these markers had a poorer prognosis compared to those that did not.”
These changes have been highlighted in yellow in the revised manuscript for ease of review.

Reviewer 2 Report
Comments and Suggestions for Authors
This is a study in which expression of a particular mRna expression is investigated and its correlation with prognosis in patients with colon cancer. It is a well designed study with adequate scientific base.with findings poorer prognosis in patients with circulating tumor cells and higher chance of liver metastasis. The authors conclude with proposing CD44v9-positive circulating tumor cells and preoperative CEA levels as prognostic markers in colon cancer.
Generally,minor English editing is required. For example in line48 "have been associated with a poor prognosis",ths "a" should be removed.
No ethical issues are found since this study is purely observetional.
The sample size seems adequate,so does the statistical analysis.
Tables are well designed and easy to understand.
References are appropriate and sychronous,no self-citations nor plagiarism are detected.
My doubt about this study is whether measuring cea and other mrna markers could change treatment methods. A comment of yours about that is needed.
Comments on the Quality of English LanguageComments are mentioned above.
Author Response
ï½¥Generally, minor English editing is required. For example in line48 "have been associated with a poor prognosis",ths "a" should be removed.
(Response)
We appreciate the reviewer`s suggestions. The English text was revised as indicated. English editing was performed by Editage (www.editage.com).
ï½¥My doubt about this study is whether measuring cea and other mrna markers could change treatment methods. A comment of yours about that is needed.
(Response)
In this study, the presence or absence of CD44v9 mRNA expression alone was considered sufficient to predict cases with poor prognosis. However, as discussed, combining the expression of CD44v9 mRNA with CEA values allowed for more detailed selection of cases with good prognosis. This selection was considered effective for identifying cases that could be excluded from postoperative adjuvant chemotherapy in stage 2 and 3 patients.

Reviewer 3 Report
Comments and Suggestions for Authors
This manuscript entitled "Presence of CD44v9-expressing cancer stem cells in circulating tumour cells and effects of CEA levels on the prognosis of colorectal cancer" by Sawai et al. studied the correlation between the expression of CD44v9 and prognosis in CRC. They found the high expression of CD44v9 correlate with worse prognosis. They also showed that CD44V9 negative or CEA level less than 5 ng/mL has better survival than those CD44V9 positive and CEA level more than 5 ng/mL. Overall, this manuscript is well described. The results are supported by the analysis of the data. However, the weakness of current manuscript lacks experimental validation to support that CD44v9 expression is critical for cell survival.
Majors:
1. The authors include two separate factors CD44v9 and CEA in their manuscript. Does the expression of CD44v9 correlate with CEA levels? The authors could perform such analysis for the intact of the story.
2. Did the authors experimentally test whether CD44v9 overexpression affect cancer cell metastasis.
3. For the survival analysis, I would advise the authors separate the cases into CD44V9-Low&CEA-Low, CD44V9-Low&CEA-High, CD44V9-High&CEA-Low, and CD44V9-Low&CEA-High groups and then do the analysis.
4. STOML2, a lipid raft protein on the cell membrane, was recently reported to regulate CRC survival (PMID: 38214751). I was wondering whether STOML2 or other factors, similarly as CEA level, worsen the CRC survival in CD44V9-high patients?
Minors:
1. Y axis labeling in Figure 2, 3,4 and 5 is not well formatted. Please revise these figures.
2. For the Supplementary Figures, the authors should organize each supplementary figure as one intact figure along with figure captions.
Author Response
Majors:
- The authors include two separate factors CD44v9 and CEA in their manuscript. Does the expression of CD44v9 correlate with CEA levels? The authors could perform such analysis for the intact of the story.
(Response)
As shown in the Results section, we have examined factors affecting the expression of CD44v9 mRNA in Tables 1 and 2, and it has been confirmed that there is no correlation with CEA in both univariate and multivariate analyses.
- Did the authors experimentally test whether CD44v9 overexpression affect cancer cell metastasis.
(Response)
We thank the reviewer for the careful review of the manuscript.
Co-author Goi used the colorectal cancer cell line HT29, which expresses CD44v9, to develop a highly metastatic cell line. By injecting these cells into the spleens of mice, they were able to create a model of liver metastasis. They reported that liver metastasis formation was inhibited when these HT29 cells were treated with an anti-CD44v9 monoclonal antibody, thus proving that CD44v9 promotes liver metastasis. Additionally, they reported that CD44v9 expression is involved in the adhesion of tumor cells to vascular endothelial cells, a critical factor for metastasis.
These changes have been added to the discussion and highlighted in yellow in the revised manuscript for ease of review.
- For the survival analysis, I would advise the authors separate the cases into CD44V9-Low&CEA-Low, CD44V9-Low&CEA-High, CD44V9-High&CEA-Low, and CD44V9-Low&CEA-High groups and then do the analysis.
(Response)
We appreciate the reviewer’s suggestions.
We added survival analysis results for groups divided into CD44V9-Low&CEA-Low, CD44V9-Low&CEA-High, CD44V9-High&CEA-Low, and CD44V9-High&CEA-High. Additionally, the survival curves are shown in Figure 5.
This change has been added to the result and highlighted in yellow in the revised manuscript for ease of review.
- STOML2, a lipid raft protein on the cell membrane, was recently reported to regulate CRC survival (PMID: 38214751). I was wondering whether STOML2 or other factors, similarly as CEA level, worsen the CRC survival in CD44V9-high patients?
(Response)
We appreciate these helpful suggestions.
We were unable to find any papers related to both STOML2 and CD44v9. However, apart from CEA, which we reported on, Sialyl Lex (SLX) has been reported as a factor affecting the prognosis of cases with high expression of CD44v9. The following statement has been added to the discussion section and highlighted in yellow in the revised manuscript for ease of review.
“In addition to CEA, which we reported on, Sialyl Lex (SLX) has been reported as a factor related to the expression of CD44v9 that affects the prognosis of colorectal cancer cases. SLX is known to play an important role in the adhesion between tumor cells and endothelial cells. It has been reported that if both are expressed in the immunostaining of colorectal cancer tissues, the prognosis is poor, whereas if neither is expressed, the prognosis is good.”
Minors:
- Y axis labeling in Figure 2, 3,4 and 5 is not well formatted. Please revise these figures.
(Response)
As suggested, We have revised the formatting of the Y-axis labels in Figures.
- For the Supplementary Figures, the authors should organize each supplementary figure as one intact figure along with figure captions.
(Response)
As suggested, We have reorganized the supplementary figures with their respective captions.
Round 2
Reviewer 3 Report
Comments and Suggestions for Authors
NA